# First Description of a Temperate Bacteriophage (vB*_FhiM*_KIRK) of *Francisella hispaniensis* Strain 3523

**DOI:** 10.3390/v13020327

**Published:** 2021-02-20

**Authors:** Kristin Köppen, Grisna I. Prensa, Kerstin Rydzewski, Hana Tlapák, Gudrun Holland, Klaus Heuner

**Affiliations:** 1Centre for Biological Threats and Special Pathogens, Cellular Interactions of Bacterial Pathogens, ZBS 2, Robert Koch Institute, 13353 Berlin, Germany; koeppenk@rki.de (K.K.); prensag@rki.de (G.I.P.); rydzewskik@rki.de (K.R.); hana.tlapak@julius-kuehn.de (H.T.); 2Centre for Biological Threats and Special Pathogens, Advanced Light and Electron Microscopy, ZBS 4, Robert Koch Institute, D-13353 Berlin, Germany; hollandg@rki.de

**Keywords:** *Francisella hispaniensis*, FhaGI-1, prophage, KIRK, bacteriophage, *Myoviridae*, vB_*FhiM*_KIRK

## Abstract

Here we present the characterization of a *Francisella* bacteriophage (vB_*FhiM*_KIRK) including the morphology, the genome sequence and the induction of the prophage. The prophage sequence (FhaGI-1) has previously been identified in *F. hispaniensis* strain 3523. UV radiation induced the prophage to assemble phage particles consisting of an icosahedral head (~52 nm in diameter), a tail of up to 97 nm in length and a mean width of 9 nm. The double stranded genome of vB_*FhiM*_KIRK contains 51 open reading frames and is 34,259 bp in length. The genotypic and phylogenetic analysis indicated that this phage seems to belong to the *Myoviridae* family of bacteriophages. Under the conditions tested here, host cell (*Francisella hispaniensis* 3523) lysis activity of KIRK was very low, and the phage particles seem to be defective for infecting new bacterial cells. Nevertheless, recombinant KIRK DNA was able to integrate site-specifically into the genome of different *Francisella* species after DNA transformation.

## 1. Introduction

*Francisella tularensis* is a Gram-negative zoonotic bacterium able to cause tularemia in a wide range of animals and in humans, where it causes various clinical expressions ranging from skin lesions to severe pneumonia, depending on the route of infection [1,2]. Infections in humans are mostly associated with the highly virulent *F. tularensis* subsp. *tularensis* (*Ftt*) and the less virulent subsp. *F. tularensis* subsp. *holarctica* (*Fth*) [3]. In individuals with compromised immune system opportunistic infections by other *Francisella* species, such as *F. novicida* (*Fno*), *F. hispaniensis* (*Fhi*) and *F. philomiragia* (*Fph*), have been reported [4,5,6]. *Fno* is an environmental, water-associated, less pathogenic species [4,5].

*Fhi* strain 3523 was isolated from a patient infected by this bacterium as a result of a cut received in brackish water in North Territory of Australia and was initially believed to be a *Fno*-like species [6]. It was the first reported *Francisella* strain from the Southern hemisphere. This strain was later re-classified to *Fhi* [6,7], a new species described in 2010 with *Fhi* type strain FhSp1 (= FSC454 = DSM 22475 = CCUG 58020), isolated from a patient in Spain [8], as the reference strain [9]. The whole genome of *Fhi* 3523 was sequenced in 2011 (Accession number CP002558) and the authors mentioned a putative prophage region together with a gene cluster coding for a putative RtxA toxin [10]. The putative prophage is not present in the genome sequence of the *Fhi* strain FSC454.

In 2013 we analyzed various *Francisella* genomes for the presence of putative functional CRISPR-Cas systems (clustered regularly interspaced short palindromic repeats/CRISPR-associated) and identified such systems in different *Fno* and *Fno*-like strains, including *Fhi* 3523 [11]. CRISPR-Cas systems are RNA-guided adaptive immunity-like systems to protect bacteria against foreign DNA, like plasmids and bacteriophages [12,13,14,15,16,17]. Parts of the foreign DNA are integrated as so called “spacer DNA” into the CRISPR system. This DNA is then used to degrade (silence) foreign DNA in a sequence-specific manner. The DNA of the CRISPR system thus represents foreign DNA which the bacterial strain has been encountered with. Analyzing the spacer sequences of the CRISPR region of various *Francisella* strains, we identified DNA sequences found entirely in the same region of the genome of *Fhi* 3523 (see Figure 1A, red arrows). We identified this region to represent a genomic island (FhaGI-1), able to generate an extrachromosomal (episomal) circular form, encoding a putative prophage [11,18]. Generation of this episomal form depend on the presence of a site-specific integrase/recombinase [11,18]. The repeat region (*attP*) and the integrase (*int*) of FhaGI-1 were then used to generate a new *Francisella* integration vector (pFIV-Val) which integrates site-specifically into the tRNA-Val gene of different *Francisella* species [19].

Prophages (integrated bacterial virus genomes) are commonly found in many bacterial genomes, encoding viruses/phages that can infect bacteria (bacteriophages). dsDNA phages can be divided in two groups: lytic and temperate (lysogenic). Lytic dsDNA phages infect bacterial cells. After replication and synthesis of new virion particles, phages are released by lysing and consequently killing their host cells. Temperate dsDNA phages are able to stably integrate into the genome of host bacteria, often at specific integration sites. However, prophages can be induced either by different environmental factors (stress conditions, temperature, UV-light, radiation) or spontaneously switch to the lytic pathway. While a lot of these prophages appear to be defective, many genes of such prophages remain functional. Thus, phage particles are produced, but the produced bacteriophages are non-functional [20]. In addition, phages are likely to serve as important vehicles for horizontal gene transfer between bacteria [21,22].

There was a preliminary communication in 1959 about lytic material derived from a culture of *F. tularensis*, yet, without describing bacteriophages [23]. In 2007, first evidence of *Legionella* bacteriophages in environmental water samples was published. One year later, another group demonstrated that bacteriophages isolated from organs of guinea pigs infected with Philadelphia 1 strain of *L. pneumophila* exhibit a certain lytic activity against *F. tularensis* [24,25]. In addition, in a PhD work in 2014 phage-like particles were described which seem to be able to infect *Francisella* cells, but neither the putative prophage genome nor the phage genome was determined and to our knowledge the data were yet not further published [26]. Considering that less is known about horizontal gene transfer in *Francisella*, and that no phage has been characterized in *Francisella* species yet [10,23,25,27,28,29], we were interested to further analyze the putative temperate bacteriophage encoded by FhaGI-1.

In this work, we could demonstrate that the prophage (FhaGI-1) of *Fhi* 3523 encodes a bacteriophage (vB_*FhiM*_KIRK) that can be induced by UV-radiation.

## 2. Materials and Methods

### 2.1. Strains, Media and Growth Conditions

Strains used in this study were *Fhi* strain 3523 (= *Fno*-like clinical isolate 3523= *Fhi* AS02-814 [CDC accession number]; kindly provided by Jeannine Petersen, CDC) [6], *Francisella* sp. strain W12-1067 (*F*-W12); [30], *Fth* LVS (ATCC 29684), *Fno* strain U112 (ATCC 15482) and *Fno* strain Fx1 (FSC156, [4], *Fph* 25015 (ATCC 25015), *Fhi* (DSM 22475); *Legionella pneumophila* strain Corby [31] and Paris (CIP 107629), *L. micdadei* (ATCC 33218), *L. dumoffii* (ATCC 33279, *L. bozemanii* (ATCC 33217), *L. oakridgensis* (ATCC 33761), and *Escherichia coli* (DH10B) One Shot^®^ TOP 10 (Invitrogen, Karlsruhe, Germany).

*Francisella* strains were cultivated at 37 °C in medium T (MT) (1% brain heart infusion broth (Difco Laboratories, Inc., Sparks, MD, USA), 1% bacto tryptone (Difco Laboratories, Inc., Sparks, MD, USA), 1% technical casamino acids (Difco Laboratories, Inc., Sparks, MD, USA), 0.005% of MgSO_4_, 0.01% FeSO_4_, 0.12% sodium citrate, 0.02% KCl, 0.04% K_2_HPO_4_, 0.06% L-cysteine and 1.5% glucose) [32,33] or on MT agar plates supplemented with hemoglobin and charcoal (MTKH plates, [19]). *E. coli* strains were grown at 37 °C in lysogeny broth (LB; 1% bacto tryptone, 0.5% yeast extract, 0.5% NaCl) or on LB agar (LB supplemented with 1.2% agar). *Legionella* strains were cultivated at 37 °C in YEB (Yeast-Extract-Broth, 1% N-(2-acetamido)-2-aminoethanesulfonic acid (ACES), 1% yeast extract, 0.04% L-cysteine, 0.025% ferric pyrophosphate) medium or on ACES-buffered charcoal-yeast extract (BCYE) agar (YEB supplemented with 1.5% agar) [34]. Kanamycin was used at a concentration of 12 µg mL^−1^ for *Francisella* and 40 µg mL^−1^ for *E. coli*; chloramphenicol was used at a concentration of 10 µg mL^−1^ for *Francisella* and 40 µg mL^−1^ for *E. coli*. 

Growth of bacteria (culture density) was monitored with Cell Growth Quantifier (CGQ, Aquila BioLabs, Baesweiler, Germany) at 37 °C and 250 rpm.

### 2.2. DNA Techniques and PCR Analysis

Plasmid DNA was prepared using the Invisorb Plasmid Mini Two Kit (Stratec, Berlin, Germany), the episomal form of the prophage was extracted using the GeneJET Plasmid-Midiprep-Kit (ThermoFisher, Waltham, MA, USA), and preparation of total DNA was done using the Blood & Tissue kit (Qiagen, Hilden, Germany) and used as control DNA in PCR experiments. The whole DNA preparation of *Fhi* 3523 has been shown to contain the episomal form of the prophage at least in a small amount [18]. Restriction enzymes were purchased from New England BioLabs (Frankfurt a. M., Germany) and were used according to the manufacturer’s protocols.

The cloning strategy to generate a recombinant phage KIRK_rec_ is outlined in Appendix A. Briefly, the whole bacteriophage was in vitro synthesized as two fragments (5′-region [GI-1 plus GFP and a second *att*-site] and 3′-region [GI-2 plus a kanamycin resistance gene]) of the prophage (GeneCust, Boynes, France). Both constructs were electroporated into strain F-W12 and integrated into the genome by the integrase of the bacteriophage (Appendix A).

PCR was carried out using a Thermocycler TRIO-Thermoblock (Biometra, Göttingen, Germany) and the TopTaq DNA polymerase (Qiagen, Hilden, Germany) according to manufacturer instructions. Primers used in this study are listed in Appendix A and are indicated in Figure 1A. In strain *Fhi* 3523 the chromosomal tRNA-Val region was amplified with primer Fha-1/Fha-4. The amplification of the circular form of FhaGI-1 (episomal/extrachromosomal) was done with primer Fha-2/Fha-3, even though this primer combination might also amplify the linear phage DNA which is generated towards the end of the lytic cycle when concatemers (multiple copies of phage DNA) are formed, cut and packed into proheads. *attP* sites (facilitating integration into host genome of the circular prophage) and linear phage DNA termini (e.g., cos and pac, depending on the DNA packing process) are usually not identical. Chromosomal integration of the bacteriophage was shown by the primer combinations Fha-1/Fha-2 and Fha-3/Fha-4. In general, initial denaturation was performed at 94 °C for 3 min, and final extension was performed at 72 °C for 10 min. Cycling conditions comprised 35 cycles at 94 °C for 30 s, 57 °C for 1 min and 72 °C for 1 min, and ~100 ng of template DNA was used. Oligonucleotides were ordered from Eurofins MWG Operon (Ebersberg, Germany).

### 2.3. Transformation of Bacteria

Plasmid DNA was introduced into *E. coli* by thermal shock (30 min on ice, 30 s at 42 °C, 2 min on ice). After transformation *E. coli* were incubated in LB medium at 37 °C for 1 h and then plated onto selective agar. For the generation of electrocompetent *Francisella*, bacteria were grown in medium T overnight, pelleted (4500 g for 15 min) and washed twice in 0.5 M sucrose. Electroporation was performed at 2.5 kV, 600 Ω and 25 µF using a Gene Pulser system (Bio-Rad, Munich, Germany). After transformation *Francisella* were incubated in medium T for 4 h at 37 °C and then plated onto selective MTKH agar plates.

### 2.4. Phage Induction Experiments 

**Temperature stress.** An overnight culture of *Fhi* 3523 was diluted with medium T to OD_600_ = 1 mL and 7 mL aliquots were incubated at 37 °C, 42 °C or 44 °C up to 24 h. Samples were collected after 1, 3, 5, 7 and 24 h of incubation and used for PCR analysis, as described below. 

**Mitomycin C stress.** An overnight culture of *Fhi* 3523 was diluted with medium T to OD_600_ = 1 mL and 7 mL aliquots were exposed to different concentrations (0 µg/mL, 0.5 µg/mL, 1.0 µg/mL and 5.0 µg/mL) of Mitomycin C (MMC) for up to 24 h. Samples incubated without MMC were used as a control for spontaneous induction of the prophage. 

**UV stress.** Using this method, DNA of *Fhi* 3523 liquid cultures were damaged by UV radiation according to the methods described by Woods [35]. An overnight culture was adjusted to OD_600_ = 1 mL and 15 mL were centrifuged at 4500× *g* for 15 min. The pellet resuspended in ½ volumes 0.01 M MgSO_4_ and 3 mL were transferred to a small petri dish. An UV hand lamp was positioned 50 cm above the petri dish to radiate a wavelength of 254 nm for different time intervals (0, 30, 60 and 90 s). After radiation the entire volume was transferred to 12 mL medium T and incubated at 37 °C and 250 rpm for up to 24 h. After 2, 3, 4, 5, 6, 7 and 24 h of incubation samples were taken and used for further analysis (see below). 

Collected samples of treated *Fhi* 3523 cultures (temperature, MMC, UV; see above) were adjusted to OD = 1, and 50 µL aliquots were centrifuged (5000× *g*, 5 min). Supernatants were discarded and pellet resuspended in 50 µL H_2_O. After heat treatment (100 °C for 10 min) samples were pelleted (5000 g, 5 min) and 15 µL of supernatants were used as DNA template for PCR analysis using primer Fha-2/Fha-3 with 10 amplification cycles.

### 2.5. Cell Disruption and Phage Purification

**Cell disruption.** To be able to isolate phage particles from UV-induced bacterial cells (see Section 2.4), independent of phage mediated lysis, cells were disrupted by sonication. After UV-radiation, bacteria were incubated at 37 °C for 5–6 h at 250 rpm and centrifuged for 15 min at 4500× *g*. The bacterial pellet was resuspended in 2 mL PBS and cooled down on ice. Afterwards the sample was sonicated on ice using a sonicator by Bandelin with an ultrasonic pulse period of 5 × 30 s, 60% amplitude and 70% pulse operation. The samples were used for EM analysis (Section 2.6).

**Phage purification.** To isolate phages from UV-induced *Fhi* 3523, bacteria were cultivated after UV induction (see Section 2.4) for 6 h and pelleted at 4500 g for 15 min. After transferring the supernatant to a new Falcon tube, DNase and RNase were added (final concentration of 1 µg/mL each) and incubated at 37 °C for 30 min, followed by sterile filtration (0.22 µm pores (Millipore, Merck, Darmstadt, Germany). Purified (not concentrated) phage samples were used for PCR analysis, EM analysis, spot tests and infection experiments (Section 2.7).

### 2.6. Electron Microscopy (EM) 

Bacterial lysates and purified phages (see Section 2.5) were diluted with distilled water (1:10) and sedimented (5000 g, 10 min.). The resulting supernatant was adsorbed to pioloform filmed, alcian blue treated copper grids, washed five times with distilled water and stained with uranyl acetate (0.5% in distilled water) for 10 s.

To visualize the assembly of new phages within the bacterial cells thin section EM was performed. UV induced liquid cultures of bacteria were sedimented at 4500 g for 15 min, and pellet was resuspended in 5 mL fixative (2.5% glutaraldehyde + 1% formaldehyde in 0.05 M HEPES buffer). After incubation in fixative at room temperature for 2 h, with occasional inverting of the vials for optimal penetration of the fixative into the cells, bacteria were embedded in low-melting point agarose (3% in distilled water). Small blocks of agarose-embedded bacteria were post-fixed with osmium tetroxide (1% in distilled water) and uranyl acetate (2% in distilled water), dehydrated stepwise in a graded ethanol series and embedded in LR White resin (Science Services GmbH, Munich, Germany) which was polymerized at 60 °C overnight. Thin sections were prepared with an ultramicrotome (UC-T; Leica, Wetzlar, Germany) and counterstained with uranyl acetate and lead citrate. 

Samples were examined using a transmission electron microscope (Tecnai Spirit, Thermo Fischer/FEI, Hennigsdorf, Germany) operated at 120 kV. Images were recorded using a charge-coupled-device camera (Megaview III, OSIS, Klosterneuburg, Austria) at a resolution of 1376 × 1032 pixel.

### 2.7. Phage Plate (Spot) Test and Phage Infection Assays

For the determination of the lysis spectrum of purified phages the soft-agar overlay technique (spot test) was used as described by Hockett and colleagues [36]. Briefly, different *Francisella* (*Fhi, Fth, Fph, F*-W12) and *Legionella* (*L. pneumophila, L. micdadei, L. oakridgensis, L. bozemanii, L. dumoffii*) species were tested as hosts. Therefore, 100 µL of exponentially grown bacteria were transferred to 4 mL of 50–60 °C heated 0.5% (*w/v*) soft-agar, gently mixed and poured over a MTKH (*Francisella* sp.) or BCYE (*Legionella* sp.) agar plate. After 30 min of curing, 10 µL aliquots of purified phage samples (see Section 2.5) were pipetted on the soft-agar, left to dry and incubated at 37 °C until a bacterial lawn became visible. The plates were then checked for lysed zones. As a negative control, aliquots of medium T-MgSO_4_ suspension and supernatants of *Francisella* strains lacking KIRK were used.

**Phage infection assays using kanamycin gene tagged KIRK (KIRK_rec_).** To further investigate the ability of KIRK to infect bacteria, we used the kanamycin resistant KIRK_rec_ phage. To isolate KIRK_rec_, bacterial strains containing the recombinant phages were induced by UV radiation to isolate the recombinant phage particles as described above (see Section 2.4 and Section 2.5). Respective bacterial strains (800 µL) were mixed with 200 µL of KIRK_rec_ containing crude phages and incubated at 37 °C for 2 h. Different dilutions were plated onto Km-containing MTKH (*Francisella* sp.) or BCYE (*Legionella* sp.) agar plates. In addition, several subcultivation were done in medium T supplemented with Km after infection. Km resistant clones and bacterial cell pellets were analyzed for the presence of episomal and/or integrated forms of recombinant KIRK by PCR (primers Fha-2/Fha-3; Fha-1/Fha-2; Fha-3/Fha-4). 

### 2.8. Phylogenetic Analysis

Multi protein sequences (in frame amino acid sequence) of genes *fhv_0008, 0012, 0018, 0023, 0024, 0025* and *fhv_0028* of KIRK and available homologous proteins from phages (*Myoviridae*): *Escherichia* T4 (*Tevenvirinae*, AF158101), *Escherichia* 186 (*Peduovirinae*, NC_001317), *Escherichia* P2 (*Peduovirinae*, KC618326), *Vibrio* VHML (*Vhmlvirus*, NC_004456), *Vibrio* VP585 (*Vhmlvirus*, NC_027981), *Wolbachia* WO2 (MK976036), *Wolbachia* WO (MN180249), *Ralstonia* phiRSP (*Jilinvirus*, MH252365), *Pseudomonas* PPpW3 (*Jilinvirus*, NC_023006), *Enterobacter* Arya (*Jilinvirus*, NC_031048), *Escherichia* ECO-1230-10 (*Jilinvirus*, GU903191) and *Escherichia* EcoM-ep3 (*Jilinvirus*, NC_025430) (obtained from GenBank) were used for amino acid sequences alignment, using the ClustalO program in Geneious. Phylogenetic analysis (phylogenetic tree) was performed by using Geneious Prime (Geneious Tree Builder, Neighbor-Joining method, *Escherichia* T4 phage as outgroup).

## 3. Results

### 3.1. Genetic Organization and Open Reading Frames (ORFs) 

We identified a genomic island (FhaGI-1) within the genome of *Fhi* strain 3523 encoding a putative prophage [11,18] (Figure 1A). The putative prophage DNA sequence starts with the *attL* site, which is a part of the bacterial tRNA-Val gene and is located upstream of protein Fhv_0001 (= FN3523_0986 of the prophage). It ends with a site-specific integrase (*int*; Fhv_0051; FN3523_1033) located upstream of the *attR* site, which represents the 3′ end of the chromosomal tRNA-Val gene (Figure 1A). In this study, we re-analyzed the prophage DNA sequence exhibiting a length of 34,259 bp and a GC content of 33.8% (Figure 1B). We identified three additional open reading frames (ORFs Fhv_0036/37/48)-compared to the annotated and published ORFs of *Fhi* 3523 [10,11], revealing now 51 putative proteins (Figure 1). The putative *Francisella* bacteriophage was named KIRK.

In Figure 1B the overview of the circular genome sequence of KIRK is shown, containing a regulatory and a phage particle production region. The regulatory region encodes the putative regulatory proteins Cro (phage CI repressor, Fhv_0044), a CI-type protein (Pro-phage repressor, Fhv_0045) exhibiting a LexA domain and a putative CII replication protein (Fhv_0043); containing an amino acid repeat (KDNNK, 3 times). Furthermore, a putative origin of replication composed of an OriR sequence (inceptor signal for DNA replication), is found upstream of four repeating units (iterons, misc-binding sites) localized within an AT-rich region [37] (see Figure 2). The sequence possesses a putative helicase (Fhv_0042) and a putative anti-repressor protein (Fhv_0038).

We performed BLASTp and HHpred analysis with all 51 ORFs and identified 44 ORFs showing similarities to known proteins of viruses of different bacteria (*Escherichia, Erwinia, Halomonas, Pseudomonas, Ralstonia, Vibrio, Wolbachia*, with identities ranging between 23–66%) (Table 1; [38]). According to BLASTp results most of these viruses belong to the family of *Myoviridae* (represented by 20 homologues ORFs) and only few to the family of *Siphoviridae* (8 homologues ORFs; Table 1). 21 of these proteins revealed the highest concordance with proteins of *Francisella* sp. SYW-9 (GCF_008711465.1, draft genome) with identities of 30–64% (Table 1). Two hypothetical proteins (Fhv_0032/49) and two integrases (Fhv_0033/51) exhibit homologs (29–62% identity) to proteins in *Fno*, *Fph* and *F. salina*. The protein Fhv_0051 has been identified earlier to be the site-specific integrase of FhaGI-1, necessary for the integration and excision of the phage integration vector pFIV-Val and thus probably also for the prophage [19].

Phage proteins are generally less conserved and therefore often not recognizable by similarity in different virions, but some phage proteins are more conserved than others, and homology of these proteins can be recognized between phage types [20]. In the region of phage particle production (Figure 1B), we identified those ORFs which putatively encode the large (Fhv-0028) and small (Fhv-0031) subunit of the terminase, the portal protein (Fhv_0018), the tail tape measure protein (Fhv_0005), as well as the less conserved capsid protein (Fhv_0017), baseplate proteins (Fhv_0012/24/25), tail proteins (Fhv_0002/3/4/6/7/14/22/23) and tail sheath protein (Fhv_008). Proteins with regulatory/replication function (Fhv_0038/42-45, see above) and the site-specific integrase/recombinase (Fhv_0051) also seem to be less conserved (Table 1 and Figure 1).

We used 7 of the more “conserved” proteins (Fhv_0008, 12, 18, 23, 24, 25, 28) and the identified respective homologs of various bacteriophages (Table 1) to perform a phylogenetic tree analysis using these proteins as 7-loci concatenated protein sequences (Figure 3). The results demonstrated that the concatenated protein sequence of KIRK forms its own branch in a subclade together with the bacteriophages vB_*EcoM*-ECO-1230-10, vB*_EcoM*-ECO-ep3, *Enterobacter* Arya and *Pseudomonas* phiRSP, all belonging to the genus of *Jilinvirus* (*Myoviridae*). Based on the sequence analysis (BLASTp and phylogenetic tree), the identified prophage of *Fhi* 3523 might belong to the family of *Myoviridae*.

### 3.2. Prophage Induction and Phage Characterization 

Since the putative prophage encodes a LexA motif containing repressor protein [18], we investigated if the prophage could be induced to be excised from the genome, to replicate and to propagate by lysing its host cell (*Fhi* 3523). To identify potential phage induction in a simplified way, a semi-quantitative PCR analysis amplifying the prophages’ circular episomal form was established (Figure 4A). To achieve this, *Fhi* 3523 was cultivated in liquid medium T and pelleted. The supernatants were heat-treated, pelleted and analyzed by PCR analysis using the primer pair Fha2/Fha3 with 5, 10, 15 and 20 PCR amplification cycles. Primer Fha-2 and Fha-3 flank the *attP* site and therefore amplify the circular extrachromosomal form of the prophage but also the linear phage DNA might by amplified which is generated after prophage induction resulting in DNA replication by forming concatemers and production of virions. Phage DNA termini sites, like cos or pac sites depending on the packing mechanism and are usually located at different sites than the *attP* site. As shown in Figure 4A, after five PCR cycles no PCR product was observed, after 10 PCR cycles a weak band was obtained and after 15 and 20 PCR cycles, respectively, a distinct band appeared. Hence, we used the primer pair Fha2/Fha3 with 10 PCR amplification cycles to detect potential phage induction recognizable by increased band intensity compared with the non-induced control. We tested UV radiation, Mitomycin C treatment and growth at different temperatures as stress conditions for phage induction (Figure 4B–D). The experiments revealed that the prophage was inducible by UV radiation, since the excised episomal form of the prophage increased after UV radiation in a dose-dependent manner (Figure 4B, 60 and 90 s). Treatment with either Mitomycin C or growth temperatures above 37 °C did not lead to an induction of the prophage (Figure 4C,D). However, a very small amount of the episomal form was detectable even without treatment (0 s, 0 µg/mL, 37 °C), but the amount did not change during the incubation time (Figure 4B–D, first row, 2–24 h). The yield of episomal form of the phage genome increased after UV radiation until about 4 to 6 h (Figure 4E, left), whereas the yield of chromosomal DNA (chromosomal gene *Fn3523_1121*) did not change considerably (Figure 4E, right). The increased yield of phage DNA was also detected in purified phage samples (DNase/RNase treated and sterile-filtered supernatants of UV-induced *Fhi* 3523 cultures; Figure 4F, upper row), which were devoid of bacterial DNA contamination (see Figure 4F, lower row). In these samples, only genes of the bacteriophage were amplified (Figure 4G, 1 to 4), but none of the chromosomal *Fhi* 3523 genes (Figure 4G, 5 and 6).

Furthermore, 5 to 6 h after prophage induction by UV radiation, negative staining electron microscopy (EM) of bacterial lysates (Figure 5A–C) and culture supernatants revealed the presence of bacteriophage particles. Phage particles composed of an icosahedral head of 52 nm (49–58 nm) in diameter and a straight tail of 82 nm (71–97 nm) in length and 9 nm (8–9.5 nm) in width (Figure 5A–C). In addition, phage particles were detected also inside of the UV-induced bacterial cells (*Fhi* 3523) by thin section EM (Figure 5D,E), indicating phage replication, transcription and assembly within bacterial cells. Furthermore, EM images indicated that the head structures are filled with electron dense material which might correspond to DNA (Figure 5A–C). In combination with the fact that none of the chromosomal *Fhi* 3523 genes were amplified in purified phage samples, the results demonstrated that the head is filled with genomic DNA of the phage. Altogether the mentioned results and results of a phylogenetic tree analysis (see above, and Figure 3) indicated that the prophage, which is present in the genome of the *Fhi* 3523, encodes a temperate bacteriophage whose structure possesses similarities to members of *Myoviridae* or *Siphoviridae*. We named this bacteriophage “virus of Bacteria, identified in *Francisella hispaniensis*, with myovirus morphotype, named KIRK” (vB_*FhiM*_KIRK), following a recent informal guide (see discussion) [39].

Since we observed bacteriophage particles after UV radiation of *Fhi* 3523, we investigated if KIRK influences the growth behavior of treated bacterial cells in liquid medium T or inhibits the growth of other bacteria on agar plates (see Section 2.7). Here, a centrifugation step usually leading to samples with high phage titers was not practical since centrifugation of KIRK resulted in apparently defective phage particles, as observed by EM. Therefore, only the purified, but not concentrated supernatants of UV-induced *Fhi* 3523 cultures were used for liquid growth assays, spot tests and infection assays. Under the conditions tested here, lysis activity of phage KIRK was not observed, since purified phage samples did not decrease the growth of any *Francisella* or *Legionella* strain tested and further, no phage-mediated inhibition zones were observed on agar plates (tested strains: *Fhi* FSC454, *Fth* LVS, *Fno* U112 and Fx1, *Fph* ATCC 25015, F-W12, *L. pneumophila* strains Paris and Corby, *L. micdadei*, *L. dumoffii*, *L. bozemanii*, *L. oakridgensis*. Using *F*-W12 and *Fth* LVS strains in further infection experiments using KIRK or KIRK_rec_, we yet could not show a successful infection of the bacteria by the bacteriophage particles.

### 3.3. The Recombinant KIRK (KIRK_rec_) 

Since the prophage containing strain was initially unavailable, we generated an in vitro synthesized recombinant form of KIRK, named KIRK_rec_ (see Appendix A).

KIRK_rec_ is tagged by a kanamycin resistance gene cassette which can be used experimentally as a selection marker in phage infection or transformation experiments. In a KIRK_rec_ positive *F*-W12 clone, KIRK_rec_ was successfully inducible by UV radiation and negative staining EM revealed the production of phage particles (Figure 6A). Complete icosahedral heads (~57 nm in diameter, 52–60 nm) were visible, but a tail structure was not detected in the investigated samples (Figure 6A). The increased size of the head of KIRK_rec_ may be a consequence of the increased size of the phage genome (additional kanamycin resitance and *gfp* gene).

To further investigate if KIRK is infective for *Francisella* and/or is able to integrate into the genome of different *Francisella* strains (prophage state), we incubated *Francisella* strains with kanamycin-resistant KIRK_rec_ particles (purified UV-induced culture supernatants). However, when plated out on agar plates containing kanamycin we did not retrieve bacterial clones containing the recombinant bacteriophage KIRK_rec_. Then, the episomal form of KIRK_rec_ were extracted from UV-induced *F*-W12 KIRK_rec_ cultures (containing KIRK_rec_ as a prophage) and used for transformation into *Francisella* strains by electroporation. Here, we obtained *Fth* LVS and *F*-W12 clones harboring KIRK_rec,_ demonstrated by the presence of the episomal form of KIRK_rec_ as well as the site-specific chromosomal integration into tRNA-Val (Figure 6B). We did not obtain *Fno* (strains U112 and Fx1) and *Fph* (strain 25015) KIRK_rec_ positive clones.

## 4. Discussion

The phage genome of KIRK is about 34,259 bp in length, it exhibits 51 ORFs and the GC content is 33.8%, and thus slightly differs from that of the bacterial *Fhi* 3523 genome (32.3%) [10]. As in other phages, the genome is organized in modular structures, with a main cluster of structural genes (Figure 1) or proteins involved in regulation/replication. In addition, analyzing ORF43, we identified a putative origin of replication which organization is comparable with that found in the *Lambda* phage [37,40]. A circular form of KIRK was assumed earlier [18]. In the *Lambda* phage, at first few circular forms of the phage genome are produced (bi-directional replication). In a second stage, long linear concatemers are synthesized by rolling circle replication and the concatemers are cut at the cos-sites (cohesive ends) into virus-sized length by the terminase. Cutting (terminase) and transport (portal protein) of the DNA into the head structure are done concomitantly [41]. Homologs of a terminase were found in the genome of KIRK and we confirmed the presence of a circular form of the phage. However, so far, we could not identify putative cos-sites of KIRK; also, other DNA replication and packaging strategies resulting in different types of DNA termini are plausible for KIRK, which needs to be investigated further.

In this work we demonstrated, that this prophage is inducible by UV radiation and bacteriophage particles were generated (see Figure 4 and Figure 5). We called the phage vB*_FhiM*_KIRK, with KIRK as the common name. From the morphology of the observed phage particles, KIRK belongs to the order of *Caudovirales* (tailed phages with dsDNA), but EM analysis alone was not sufficient to classify KIRK as part of the *Myoviridae* or *Siphoviridae*. On the one hand a neck structure, separating tail and head, was not found and the observed tail structure seems to be too thin (9 nm, Figure 5) for phages belonging to *Myoviridae,* which usually possess neck and relatively thick double-sheathed tails (16–20 nm). The analysis of the amino acid sequence of ORF Fhv_0008 revealed the presence of Pfam-Phage_sheath (1 and 1C) domain and BLASTp analysis did not reveal deletions within the sequence or evidence for a defective sheath protein. However, the tail width of about 9 nm found by EM fits to the size of the tail tube width of the *Myoviridae* which may be exposed by incomplete assembly of the tail or by a shedding of the tail tube/sheath protein [42]. This may also explain the inability of KIRK to re-infect bacteria (see below) and thus KIRK present in *Fhi* strain 3523 may represent a defective form of the bacteriophage. On the other hand, the obtained tail structure is short (71–97 nm), and the tails of *Siphoviridae* phages are rather long [41]. In addition, it is not clear if the tail of KIRK is either contractile and rigid, or non-contractile but flexible, which are also common features to discriminate between *Myoviridae* or *Siphoviridae* phages. However, of the proteins encoded by KIRK, 20 and 8 proteins exhibit similarity to *Myoviridae* and *Siphoviridae*, respectively, according to BLAST analysis. In addition, phylogenetic tree analysis of a 7 loci concatenated protein sequence (of the most conserved proteins) corroborated KIRK belonging to the family of *Myoviridae,* since the phylogenetic closest bacteriophages all belong to the genus *Jilinvirus*, belonging to the family of *Myoviridae* (see Figure 3).

The observed induction of the prophage by UV radiation is explainable by the presence of ORF45, encoding a putative homolog of the phage repressor CI exhibiting a LexA motif. In general, DNA damaged by UV radiation activates a host defense mechanism (SOS response) that helps the bacterium to survive, by degrading the bacterial LexA repressor, which represses a set of bacterial genes involved in DNA repair. However, this SOS system also leads to the degradation of CI, and thus to the induction of the prophage [43,44]. Our results indicated that this may also be true for the induction of KIRK. The PCR results (Figure 4, control, (0 s)) indicated that a small amount of extrachromosomal KIRK is produced without induction which could be explained by the fact that temperate bacteriophages spontaneously and randomly are induced in a very small fraction of bacterial cells. Spontaneous lysis occurs once in about 10^4^ bacteria and is also RecA-dependent as a result of rare sporadic DNA damage [20]. Nevertheless, KIRK seems to be able to lyse its host (*Fhi* 3523) at least to a small extent since phage particles were observed in the supernatant of UV-induced *Fhi* 3523 cultures by electron microscopy and the yield of phage DNA increased in culture supernatants due to UV induction (Figure 4F). Basically, most phages with double-stranded nucleic acid genomes use lysozymes (endolysin) and holins for bacterial cell lysis. The lysis of the bacteria occurs at a strict defined time during the infective phage cycle [45]. So far, a holin protein has not been identified in silico in the genome of KIRK. Though, holins are a diverse group of enzymes with more than 250 proteins in more than 50 families, partially without significant sequence similarities making an in silico identification of such protein quite difficult [46]. ORF Fhv_0050 encodes a predictive endolysin according to HHpred analysis with 75% probability to *Enterobacteria* phage T4 and Fhv_0025 of KIRK encodes a putative phage baseplate protein with lysozyme activity which exhibits 42% protein identity to the gp25-like lysozyme of Enterobacteria phage P88. This protein shows similarities to the T4 phage protein 25 which is a structural component of the baseplate and has also an acidic lysozyme activity [47]. Here, further investigations are needed including analysis of protein topology and function to identify proteins involved in lysis process. Moreover, other proteins are also involved in the phage-induced process of bacterial cell lysis like antiholins and spanins which have not been identified for KIRK yet [48].

For the survival of progeny virions, phages need to initiate infection including recognition and absorption to a suitable host, penetration of the cell wall and injection of phage DNA. The interaction between phages and host cells is primary driven by phage tail proteins and bacterial receptors. Therefore, intact tails are mandatory for successful infection and the following intracellular lifecycle including lysogenic and lytic pathway. So far, we were unable to show experimentally a KIRK infection of different *Legionella* and *Francisella* species. There are several explanations for this observation: (i) The tested conditions were insufficient and need to be optimized, especially the phage purification to reach high-concentrated phage samples. We used supernatants of UV-induced *Fhi* 3523 cultures for infection experiments since centrifugation of KIRK resulted in deformed phage particles observed by EM. (ii) KIRK might poorly attach, penetrate or inject its DNA into its host due to so far unknown reasons. (iii) The phage particles of KIRK released by UV-induced *Fhi* 3523 cells lack the tail sheath and thus are unable to re-infect new host bacteria; (iv) KIRK might be a “strong” temperate phage which predominately integrates into the genome rather than entering the lytic cycle. Nevertheless, DNA of KIRK_rec_ (episomal form) transformed into *Francisella* by electroporation integrated successfully into the genome of *Fth* LVS and *F*-W12 demonstrating the prophage state of KIRK_rec_ (Figure 6B). Moreover, the episomal form of the prophage was observed in *Fth* LVS and *F*-W12 after transformation. However, no *Fno* KIRK_rec_ positive clone was obtained, suggesting that the active CRISPR-Cas system of *Fno*, containing KIRK-specific spacer DNAs, acts actively against the invading KIRK DNA [11,49].

Phages like the hereby described KIRK are likely vehicles for horizontal gene transfer and environmental phages are implicated in the network of genetic exchange among bacteria, which is also involved in the evolution of pathogens [50]. Recently, a putative *Francisella* conjugative element has been described but it is unknown if it is really involved in horizontal gene transfer [28]. However, since horizontal gene transfer in *Francisella* is not well understood, it is important to document and characterize putative gene transfer by temperate bacteriophages of *Francisella*. In addition, since phages are able to lyse bacteria, bacteriophages can be used also to specifically kill bacteria. Bacteriophage treatment is used as an alternative therapy for human infections, in food production and processing, as well as for microbial decontamination [51,52,53]. Although our phage is lysogenic and lytic capacity is not well understood, it possesses the potential usage in the abatement of *Francisella* sp. e.g., by genetically enhancing the lytic capacity and elimination of the lysogenic modus of the phage. Such a modified bacteriophage KIRK might be helpful to decontaminate environmental areas exhibiting high concentrations of the highly pathogenic species *F. tularensis*.

**In summary**, KIRK seems to be a temperate bacteriophage present in the environment, and the prophage KIRK present in strain *Fhi* 3523 seems to represent a defective form of this bacteriophage, demonstrated by following observations: (i) the identification of the complete genome of the prophage in a *Francisella* strain (*Fhi* 3523), indicating that *Fhi* 3523 has been infected successfully by KIRK in its natural habitat, (ii) the identification of various different anti-KIRK spacer sequences in the CRISPR region of different *Fno* strains [11], indicating a direct contact of these strains with the bacteriophage KIRK in the environment, (iii) the demonstration of phage assembly and multiplication within bacterial cells (Figure 5 D,E) and (iv) the detection of phage particles in bacterial culture supernatants indicating lytic capacity. Furthermore, KIRK may be ubiquitous in natural (aquatic) habitats, since KIRK or KIRK-specific DNA spacers (CRISPR) were found in strains isolated in Australia (*Fhi* 3523, from a patient infected by brackish water) and in different parts of the USA (*Fno* U112, Utah, aquatic environmental isolate; *Fno* GA99-3548, Louisiana; GA99-3549, California, *Fno* Fx1, Texas, patient isolate). The prophage present in strain *Fhi* 3523 may be unable to re-infect new bacteria because of a defective tail sheath structure.

## Figures and Tables

**Figure 1 viruses-13-00327-f001:**
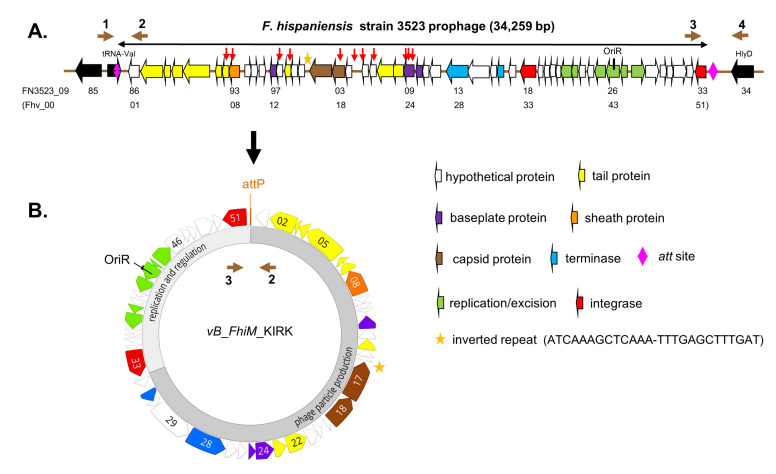
Organization of *Francisella* bacteriophage vB_*Fhi*M_KIRK as prophage (**A**) or its episomal form (**B**)**.** (**A**)**:** The prophage is integrated within the tRNA-Val gene (pink arrow). The *att* sites (*attL* and *attR*) are indicated by a pink trapezium. The *attR* site corresponds to the 3′ end of the tRNA-Val. Chromosomal genes are given in black and genes of the bacteriophage are given in different colours according to their respective putative function based on BLASTp analysis (see also Table 1). A putative origin of replication (OriR) is indicated. Gene numbers are indicated below the genes as published for *Fhi* 3523 (FN3523; CP002558) or as determined in this work for the bacteriophage KIRK (FhV_0001 to FhV_0051). Location of the spacer DNAs identified in the CRISPR-Cas systems of different *Francisella* strains are indicated above of the genes by red arrows (Schunder et al., 2013; modified). Primer used in this study are indicated as brown arrows (for details see text). (**B**)**:** Gene organization of the episomal form of bacteriophage KIRK are given in different colours (see (**A**)) due to their putative function and are clustered in “replication and regulation” and “phage particle production”. The site-specific DNA region of KIRK (*attP*) responsible for the integration into the genome of host cells (*attB*, not shown) is indicated.

**Figure 2 viruses-13-00327-f002:**
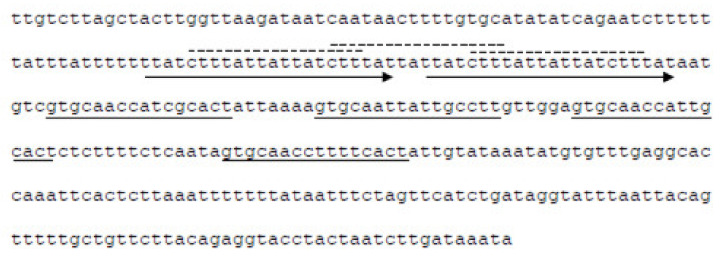
The putative origin of replication of vB_*FhiM*_KIRK. The origin is composed of the putative inceptor signal for DNA replication (dotted lines, three times), found within a direct repeat sequence (arrows), and four misc-binding sites (underlined sequence). The region is localized within gene *fhv_0043* (see Figure 1).

**Figure 3 viruses-13-00327-f003:**
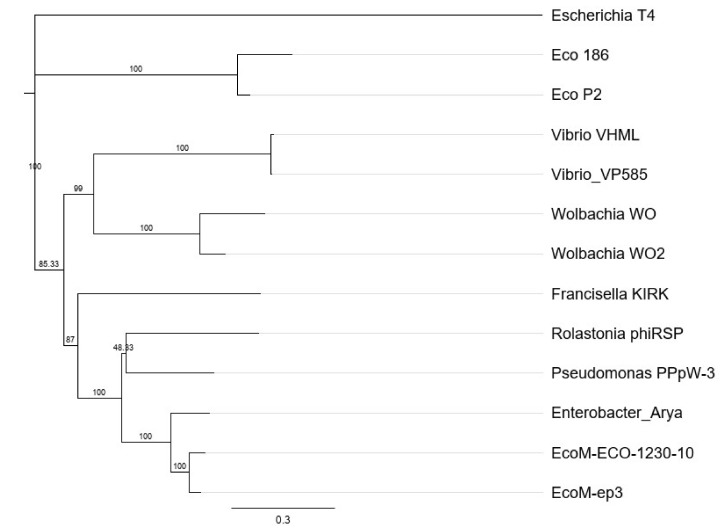
Phylogenetic tree analysis of the bacteriophage KIRK. 7 loci concatenated protein sequence of genes *fhv_0018, 0024, 0012, 0008, 0025, 0028, 0023*) and available homologous proteins from bacteriophages *Escherichia* T4 (*Tevenvirinae*, AF158101), *Escherichia* 186 (*Peduovirinae*, NC_001317), *Escherichia* P2 (*Peduovirinae*, KC618326), *Vibrio* VHML (*Vhmlvirus*, NC_004456), *Vibrio* VP585 (*Vhmlvirus*, NC_027981), *Wolbachia* WO2 (MK976036), *Wolbachia* WO (MN180249), *Ralstonia* phiRSP (*Jilinvirus*, MH252365), *Pseudomonas* PPpW3 (*Jilinvirus*, NC_023006), *Enterobacter* Arya (*Jilinvirus*, NC_031048), *Escherichia* ECO-1230-10 (*Jilinvirus*, GU903191) and *Escherichia* EcoM-ep3 (*Jilinvirus*, NC_025430) were used for amino acid sequences alignment using the ClustalO program in Geneious. The phylogenetic tree was generated by using Geneious Tree Builder, Neighbor-Joining method and *Escherichia* T4 phage as outgroup.

**Figure 4 viruses-13-00327-f004:**
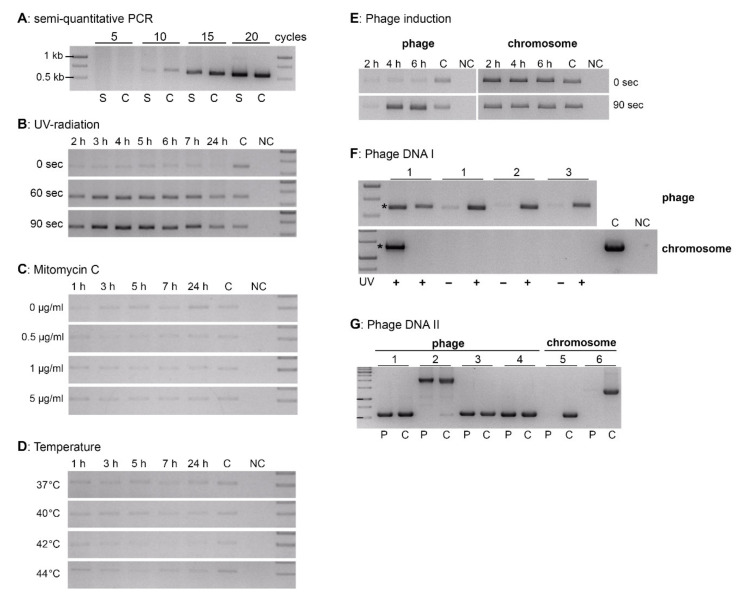
Bacteriophage induction. (**A**): Semi-quantitative PCR analyses. Supernatants of *Fhi* 3523 lysates (S) and control DNA of *Fhi* 3523 (C) were analyzed using primer Fha-2/Fha-3 (product size: 538 bp) with 5, 10, 15 and 20 PCR amplification cycles. (**B**–**D**): *Fhi* 3523 was exposed to UV radiation at 254 nm for 0, 60 and 90 s (**B**); treated with 0, 0.5, 1 and 5 µg/mL mitomycin C (**C**); or incubated at 37, 40, 42 and 44 °C (**D**). Samples were collected after various time points post-induction and used for PCR analyses with Fha-2/Fha-3 and 10 amplification PCR cycles. (**E**): PCR analyses were performed targeting the phage KIRK using primers Fha-2/Fha-3 (left, 538 bp) and bacterial genome of *Fhi* 3523 with primers Fhis_R13/Fhis_U13A (right, 1289 bp) after 0 and 90 s of UV radiation (254 nm) at different time points. (**F**): Supernatants (1, 2, 3, three replicates) of *Fhi* 3523 cultures treated with (+) or without (−) UV for 90 s at 254 nm were analyzed by PCR detecting the phage (Fha-2/Fha-3, 10 PCR cycles, upper row) and the bacterial *Fhi* 3523 chromosome (Fhis_U13A/Fhis_R13, lower row). All supernatants were treated with DNase and RNase and sterile filtered prior using in PCR analyses, except for sample marked by asterisk (control) which was not treated with DNase and RNase. (**G**): Purified, UV-induced phage samples (P) and control DNA of *Fhi* 3523 (C) were analyzed targeting FhaGI-1 (PCR 1–4) and chromosomal regions of *Fhi* 3523 (PCR 5, 6), respectively. 1: Fha-2/Fha-3 (538 bp); 2: Fha996_U/Fha997_R (2024 bp); 3: F1_out_U/F2_out_R (554 bp); 4: F2_out_U/F3_out_R (530 bp); 5: Fha-1/Fha-4 (617 bp); 6: Fhis_R13/Fhis_U13A (1289 bp). C = control DNA of *Fhi* 3523 (whole DNA of bacterial cell lysates, see Section 2.2); NC = no template control; DNA ladder 1 kb GeneRuler was used.

**Figure 5 viruses-13-00327-f005:**
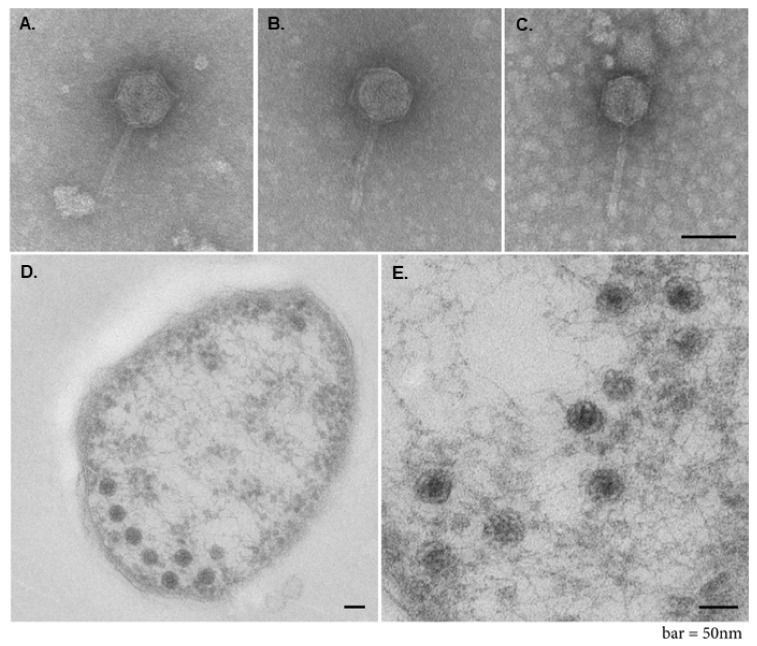
Electron micrographs of *Francisella* phage vB_*FhiM*_KIRK. (**A**–**C**): Transmission EM of purified KIRK samples (see Section 2.5) stained with uranyl acetate. Phage particles are composed of an icosahedral head ~52 nm in diameter and a tail structure of ~82 nm in length and ~9 nm in width. (**D**,**E**): Thin section EM of UV treated *Fhi* 3523 cells, showing multiple phage particles inside of one bacterial cell. Bars = 50 nm.

**Figure 6 viruses-13-00327-f006:**
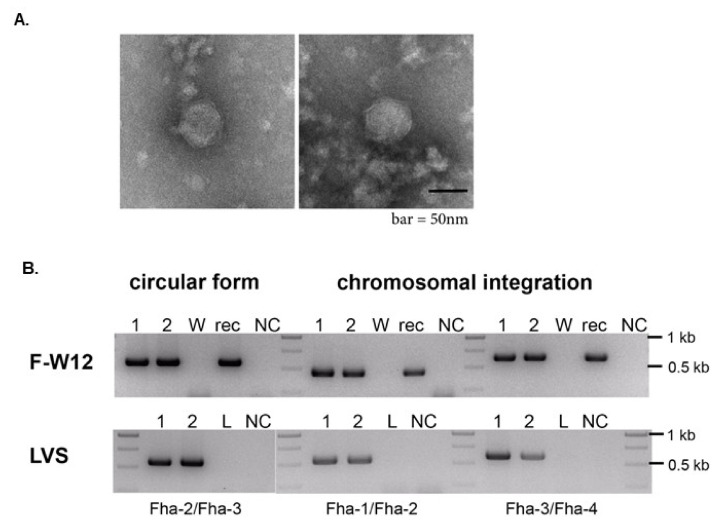
The recombinant bacteriophage KIRK_rec_. (**A**): Transmission EM of purified KIRK_rec_ stained with uranyl acetate. Phage particles are composed of an icosahedral head ~57 nm. (**B**): Infection of *Francisella* with KIRK_rec_. *Fth* LVS and *F-*W12 were transformed with the episomal form of KIRK_rec_ by electroporation. Obtained clones were tested regarding the circular form of KIRK_rec_ using primer Fha-2/Fha-3 and the chromosomal integration using Fha-1/Fha-2 and Fha-3/Fha-4, here Fha-1 and Fha-4 are species-specific primers binding in the genome of *F*-W12 (Fha-1^W12^, Fha-4^W12^) and *Fth* LVS (Fha-1, Fha-4*), respectively, see Appendix A. 1, 2: *F*-W12 and *Fth* LVS KIKR_rec_ clones, respectively; W: chromosomal DNA of *F*-W12, rec: *F*-W12 KIRK_rec_ clone which was obtained by in vitro synthesis and cloning (see Appendix A), and was used for extraction of circular form of KIRK_rec_; NC: no template control; L: chromosomal DNA of *Fth* LVS.

**Table 1 viruses-13-00327-t001:** Putative proteins encoded by ORFs Fhv_0001 to Fhv_0051 of KIRK.

ORF Number	Aa	Motif/Putative Function	putative Function in Phages According to HHpred # (Probability)	BLASTp Viruses (Identity)	BLASTp *Francisella* Group (Identity *)
Fhv_0001 (0986) **	197	2× internal repeat, HP	Major capsid protein (50%)	*Aedes pseudoscutellaris* reovirus VP6 (25%)	-
Fhv_0002 (0987)	408	Phage_GPD, GpD phage late control protein D	Tail protein (100%) °	*Wolbachia* phage WO (30%) °	*Francisella* sp. SYW-9 (47%)
Fhv_0003 (0988)	066	Phage_tail_X Phage tail protein	-	*Wolbachia* phage WO (55%) °	*Francisella* sp. SYW-9 (55%)
Fhv_0004 (0989)	138	Phage_P2_GpU Phage tail assembly	Major tube protein gp53 (86%)	*Caudovirales* phage (29%) °	*Francisella* sp. SYW-9 (43%)
Fhv_0005 (0990)	607	Coiled coil, 3× LCRs Phage tail tape measure protein	Tape Measure Protein gp57 (99%)	*Vibrio* phage VpKK5 (36%)	*Francisella* sp. SYW-9 (36%)
Fhv_0006 (0991)	089	Phage_TAC_7 Phage tail assembly chaperone, Myoviridae	Lambda integrase (48%) DNA-binding protein gp33 (41%) Tail assembly chaperone (38%)	*Pseudomonas* phage PPpW-3 (31%) °	*Francisella* sp. SYW-9 (53%)
Fhv_0007 (0992)	159	Phage_tube Major tail tube protein	Tail tube protein gp19 (80%) °	*Wolbachia* phage WO (35%) °	*Francisella* sp. SYW-9 (51%)
Fhv_0008 (0993)	385	Phae_sheath_1, Phage_sheath_1C Phage tail sheath protein	Tail sheath protein Gp18 (100%) °	*Wolbachia* phage WO (49%) °	*Francisella* sp. SYW-9 (56%)
Fhv_0009 (0994)	106	HP	Integrase (21%)	-	-
Fhv_0010 (0995)	081	HP	Gene 9 protein Knob (23%)	-	-
Fhv_0011 (0996)	066	LCR,	arc repressor (39%)	*Pseudomonas* phage EL (50%) °	*F. tularensis* subsp. *novicida* PA10-7858 (50%)
Fhv_0012 (0997)	198	Phage_base_V, Baseplate assembly protein V	Baseplate assembly protein V (100%) °	*Wolbachia* phage WO (34%) °	*Francisella* sp. SYW-9 (36%)
Fhv_0013 (0998)	163	HP	Minor tail protein U (97%)	*Wolbachia* phage WO (26%) °	-
Fhv_0014 (0999)	166	166 aa, LCR, minor tail_Z superfamily Phage minor tail protein	-	*Halomonas* virus HAP1 (26%) °	*Francisella* sp. SYW-9 (40%)
Fhv_0015 (1000)	111	HP	Tail attachment protein (95%)	-	*Francisella* sp. SYW-9 (30%)
Fhv_0016 (1001)	063	HP	-	Mediterranean phage uvMED (36%)	-
Fhv_0017 (1002)	600	Peptidase_S78, Phage_capsid Major phage capsid protein	Major capsid protein (100%)	*Escherichia* phage vB_EcoM_ECO1230-10 (37%) °	*Francisella* sp. SYW-9 (40%)
Fhv_0018 (1003)	473	Phage_portal_2, portal_lambda Phage hole protein, forming DNA-ejection hole	Portal protein (100%)	*Enterobacter* phage Arya (34%) °	*Francisella* sp. SYW-9 (39%)
Fhv_0019 (1004)	088	HP, coiled coil	Head-to-tail joining protein W (89%)	-	-
Fhv_0020 (1005)	110	DUF1353, conserved HP	-	*Fusobacterium* phage Funu2 (36%) °	*F. marina* (46%)
Fhv_0021 (1006)	183	HP, DUF4376	Tail fiber assembly protein U (79%) °	-	-
Fhv_0022 (1007)	324	DUF3751, Pfam_12571 Phage tail fibre protein	Long-tail fiber proximal subunit (75%) °	*Salmonella* phage vB_SnwM_CGG4-1 (35%) °	*Francisella* sp. SYW-9 (42%)
Fhv_0023 (1008)	195	Tail_P2_I, Phage tail protein I	Baseplate wedge protein gp6 (77%) °	*Ralstonia* phage phiRSP (31%) °	*Francisella* sp. SYW-9 (47%)
Fhv_0024 (1009)	281	Baseplate_J, (P2 phage), gpJ Phage-related baseplate assembly protein	Baseplate wedge protein gp6 (100%) °	*Pseudomonas* phage PPpW-3 (36%) °	*Francisella* sp. SYW-9 (52%)
Fhv_0025 (1010)	112	GPW_gp25 T4 phage, V1_zyme Phage baseplate protein, lysozyme activity	Baseplate wedge protein gp25 (100%) °	*Ralstonia* phage phiRSP (48%) °	*Francisella* sp. SYW-9 (53%) *F. philomiragia* (37%)
Fhv_0026 (1011)	131	HP, coiled coil, TM	Fibritin (56%) °	-	-
Fhv_0027 (1012)	169	HP, LCR, TM	-	-	*F. marina* (47%)
Fhv_0028 (1013)	602	Terminase_GpA, Phage terminase large subunit, DNA packaging	Terminase (100%)	*Enterobacter* phage Arya (48%) °	*Francisella* sp. SYW-9 (59%)
Fhv_0029 (1014)	671	HP, 6x LCR	-	-	-
Fhv_0030 (1015)	093	HP	-	-	*F. marina* (40%)
Fhv_0031 (1016)	172	Phage_Nu1 SF Minor subunit Nu1 of terminase	Regulatory protein cox (98%) °	Mediterranean phage uvMED (29%)	*F. philomiragia* (57%) *F. novicida* (57%) *F. salina* (57%)
Fhv_0032 (1017)	128	HP	Middle operon regulator (100%) °	-	*Francisella novicida* (33%)
Fhv_0033 (1018)	422	Arm-DNA-bind_3, Phage_Int_P4, Phage integrase	Integrase (100%)	*Pseudomonas* phage phiAH14b (32%)	*F. salina* (54%) *F. novicida* (54%) *F. philomiragia* (56%) *Francisella* sp. *SYW-9* (48%)
Fhv_0034 (1019)	106	HP, LCR	-	-	-
Fhv_0035 (1020)	072	HP	-	-	*F. marina* (70%)
Fhv_0036 (n.a.)	061	HP	-	*Megaviridae* environmental sample (41%)	*Francisella* sp. FSC1006 (43%) *F. marina* (45%)
Fhv_0037 (n.a.)	075	HP, TM, conju_TIGR03752, Integrating conjugative element protein	Fusion of phage phi29 Gp7 protein and Cell division protein FtsB (64%)	-	*Francisella* sp. SYW-9 (50%)
Fhv_0038 (1021)	240	Phage_pRha, ANT Phage regulatory and anti-repressor protein	Anti-sigma effector (45%)	*Lactobacillus* phage phiEF-1.1 (66%)	*Francisella* sp. SYW-9 (54%) *F. marina* (71%)
Fhv_0039 (1022)	120	Phage_TIGR01671, YopX Putative phage protein	HP ORF041 (*Staphylococcus* phage, 100%)	*Clostridium* phage phiCT19406C (41%)	-
Fhv_0040 (1023)	100	HP, coiled coil	Long tail fiber distal subunit (24%) °	-	-
Fhv_0041 (1024)	085	HP	Regulatory protein cox (99%)	-	*Francisella* sp. SYW-9 (64%) *F. philomiragia* (48%)
Fhv_0042 (1025)	212	Inhibitor_G39P Blocking G40P replicative helicase	Replisome organizer (100%) °	-	*Francisella* sp. FSC1006 (46%) *F. philomiragia* (34%)
Fhv_0043 (1026)	240	Phg_2220_C, (internal repeat) Phage replication protein	DNA-binding protein TubR (97%)	*Lactobacillus* prophage Lj771 (50%) °	*Francisella* sp. FSC1006 (57%)
Fhv_0044 (1027)	056	P22_Cro (lytic growth), Cro protein, phage_CI_repressor	Repressor protein (98%)	*Erwinia* phage vB_EhrS_59 (45%)	*Francisella* sp. FSC1006 (45%)
Fhv_0045 (1028)	265	Peptidase_S24_S26 SF, CI-type HTH_XRE domain, Pro-phage repressor (CI), LexA protein domain,	Lambda Repressor (100%)	*Streptococcus* phage PH15 (48%)	*F. philomiragia* (54%) *F. salina* (52%) *F. marina* (51%) *F. novicida* (57%)
Fhv_0046 (1029)	293	HP	-	*Marinobacter* phage AS1 (23%)	-
Fhv_0047 (1030)	112	HP, DUF4325	-	*Campylobacter* phage CP30A (27%) °	-
Fhv_0048(n.a.)	125	HP, TM	-	-	-
Fhv_0049 (1031)	264	HP	-	-	*F. philomiragia* (45%) *F. noatunensis* (45%)
Fhv_0050 (1032)	130	TM, bPH_2 Putative transmembrane protein	Endolysin (75%) °	*Serratia* phage phiMAM1 (39%)	*Francisella* sp. W12-1067 (46%)
Fhv_0051 (1033)	375	Phage_integrase, Arm-DNA-bind_3 Site-spezific integrase/recombinase	Integrase (100%)	Prokaryotic dsDNA virus sp. (30%)	*F. adeliensis* (62%) *F. philomiragia* (56%) *F. marina* (52%) *F. salina* (51% )*Francisella* sp. SYW-9 (49%)

* protein identity >30%; ** ORF number (FN3523_00xx) in the genome of *F. novicida*-like strain 3523 (CP002558.1), now renamed to *F. hispaniensis*; # https://toolkit.tuebingen.mpg.de (accessed on 12 February 2021) [38]; ° *Myoviridae* bacteriophage; LCR, low complexity region; DUF, domain of unknown function; HP, hypothetical protein; TM, transmembrane region; SF, super family; n.a., not annotated.

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
