# Peer review of "First Description of a Temperate Bacteriophage (vB_FhiM_KIRK) of Francisella hispaniensis Strain 3523"

_viruses, 2021, doi:10.3390/v13020327_

Round 1

Reviewer 1 Report

The revised paper from Köppen et al has greatly improved, and has taken into consideration nearly all of my comments. There is four minor points that I would like to be revised:

In the abstract, the authors state : “The morphotypic and phylogenetic analysis indicated that this phage seems to belong to the Myoviridae family of bacteriophages.”  I would change that to “The genotypic and phylogenetic analysis indicated that this phage seems to belong to the Myoviridae family of bacteriophages.”, as it is discussed in the paper that the tail is too thin to be a contractile tail.

Figure 1-Table 1, here was my previous comment: “It would help if in figure 1, the tail proteins were labelled with the proposed function based on the data presented in Table 2. Gene organisation in the structural gene region is very conserved among phages, and it would help to be able to see the organisation of genes in KIRK. (By the way, Table 2 was not in the manuscript but presented as unpublished data, why? Also, the legend in Fig 1B is far too small. “Replication and regulation” and “phage particle production” are barely readable, and the gene numbers are truncated!) To identify protein function, did the authors consider using the powerful tool HHPRED? It is very easy to use and extremely powerful to find homologues based on the secondary structure prediction rather than sequence similarity. I highly recommend it.”   The figure is now clearer, but I feel the authors didn’t take fully advantage of the HHPRED results, to have a better idea of the function of the tail proteins in particular. For example, Fhv_0004 and Fhv_0007 both point to “Major tube protein”, but HHPRED proposes gp53 and gp19. Do these refer to Phage T4? More specifically, to which proteins are they close to? Baseplate? They can’t both be Major Tube proteins.  I had a look: if they refer to T4, then gp53 is indeed a Myo-baseplate sheath protein, and gp19 is the Major Tube Protein. This gives now more information and completes the genomic analysis. Please, do that exercise for all proteins, and note the rotative role of proteins on the figure.

To continue in that direction, the authors have long discussion about the absence of identified holins and endolysins… However, HHPRED now predicts the Fhv_0050 as an endolysin!!! This should be included in the discussion!!!

In the legend of Fig. 1, the authors refer to orange arrow and orange trapezium, which I see pink. They also refer to a black star, which I see orange… Also, panels A and B are not indicated on the figure.

Author Response

Thank you for your suggestions.

As demanded by Reviewer 2,

-1. “morphotypic” has been changed to “genotypic” (Abstract),

-2. As stated in the manuscript (lines 531 to 535; word-doc), in our opinion, a deeper analysis of the predicted function of BLAST results and HHPRED (sometimes not the same results) should be done in a further deeper characterization of this bacteriophage and should be part of a following manuscript.  

-3. We introduced this result in the text body (lines 526-527)

-4. As suggested by the reviewer, we corrected the Figure 1 (added “A:” and “B:”) and its legend

Reviewer 2 Report

Koppen et al characterized a bacteriophage (vB_FhiM_KIRK) which had been derived from a Francisella strain.  Characterization included deriving the methods of induction, EM studies, and sequence analysis. 

The quality of the manuscript is sound and the authors appear to have revised as needed by the previous reviewers.

Overall, the study appears sounds and for this revised manuscript, the authors addressed the concerns of the previous reviewers and made the necessary revisions.

Author Response

Thank you very much for your kind words and your positive decision.

This manuscript is a resubmission of an earlier submission. The following is a list of the peer review reports and author responses from that submission.

Round 1

Reviewer 1 Report

The question of whether functioning phages are present in the genome of .F. hispaniensis is of interest but not entirely novel, it has been addressed in at least one phD thesis from 2014 (1). Assuming that the authors are aware of said dissertation it would seem appropriate for it to be mentioned. The authors claim to have isolated and characterized the first known functioning temperate phage from Francisella, disregarding the issue of whether it is the first isolation or not, the authors gives no data supporting the claim that the isolated phage is either functional or temperate. Rather, they present data that correctly interpreted strongly indicates that the phage is defective and hence should be indicated as a degraded prophage.

The induction experiments fails to indicate/support the presence of an infectious phage. The experiments indicate the presence of an episomal form of the prophage that can build some form of phage particles but is non-infectious. To elucidate whether the produced phage particles are infectious or not, an experiment should be performed where F. hispaniensis 3523 is cured of the prophage and then re-infected with the phage particles. After allowing for infection, lysis and integration of the phage genome into the host genome should be monitored. If either event occurs then the phage is a functional phage.

As for the way the manuscript is written it is hard to distinguish whether the reasoning and thinking in the manuscript originates from the manuscript data or from data already published in other publications by the same group (2, 3), hence the scientific contribution of this manuscript is minimal at best.

1: Isolation and characterization of a novel bacteriophage, asc10, that lyses francisella tularensis
https://mountainscholar.org/bitstream/handle/10217/88407/Alharby_colostate_0053A_12804.pdf?sequence=1

2: First indication for a functional CRISPR/Cas system in Francisella tularensis
https://www.sciencedirect.com/science/article/pii/S1438422112000902?via%3Dihub

3: Identification and characterization of episomal forms of integrative genomic islands in the genus Francisella
https://www.sciencedirect.com/science/article/pii/S1438422115300060

Author Response

The question of whether functioning phages are present in the genome of .F. hispaniensis is of interest but not entirely novel, it has been addressed in at least one phD thesis from 2014 (1). Assuming that the authors are aware of said dissertation it would seem appropriate for it to be mentioned.

R:We now stated in the introduction: “In addition, in a PhD work in 2014 phage-like particles were described which seem to be able to infect Francisella cells, but neither the putative prophage genome nor the phage genome were determined and to our knowledge the data were yet not further published (26)”. However, used strain in the thesis were Fno strains (U112), but not Fhi 3523 (Fno-like strain 3523), the EM analysis (pictures) is not convincing, phage encoding DNA was not verified and the mentioned DNA region (putative prophage) did not correspond to the head size of the phage particle described.

The authors claim to have isolated and characterized the first known functioning temperate phage from Francisella, disregarding the issue of whether it is the first isolation or not, the authors gives no data supporting the claim that the isolated phage is either functional or temperate. Rather, they present data that correctly interpreted strongly indicates that the phage is defective and hence should be indicated as a degraded prophage.

R:We now stated that the version of KIRK present in Fhi 3523 may be a defective phage (no re-infection detected), but we clearly showed that the prophage is inducible and that the phage DNA integrates site-specific into the genome of Fth LVS and F. sp. strain W12-1067.

The induction experiments fails to indicate/support the presence of an infectious phage. The experiments indicate the presence of an episomal form of the prophage that can build some form of phage particles but is non-infectious. To elucidate whether the produced phage particles are infectious or not, an experiment should be performed where F. hispaniensis 3523 is cured of the prophage and then re-infected with the phage particles. After allowing for infection, lysis and integration of the phage genome into the host genome should be monitored. If either event occurs then the phage is a functional phage.

R: See comment above

As for the way the manuscript is written it is hard to distinguish whether the reasoning and thinking in the manuscript originates from the manuscript data or from data already published in other publications by the same group (2, 3), hence the scientific contribution of this manuscript is minimal at best.

R:We do not agree with reviewer 1. See also comments of reviewer 2.

Reviewer 2 Report

The paper from Köppen et al describes the thorough characterisation of a prophage that can be induced to form bacteriophage particles that are released  by UV light, from a Francisella bacterial strain. This is the first phage described for these bacterial species, opening the way to using it, after optimisation, as a means to fight against the pathogenis species F. tularensis.

This work is clearly important. The experiments are well planned, performed carefully and well illustrated. The authors cite the literature adequately, and give credit to related research. However, I have a few concerns about the interpretation of some of the experiments.

The authors conclude that phage KIRK, when induced by UV irradiation, is a biological active phage. I would not be so conclusive, as it could not be shown that the particle could indeed infect, yet induce lysis of bacteria. The authors very rightly conclude that KIRK might be a myophage, but the EM images do not support this idea, the tail being too thin. I completely support this conclusion, and would go further in the analysis: from the genetic/phylogenetic analysis, the prophage seems to be coding for a contractile tail. The produced particles clearly lack the sheath around the tail tube (a diameter of 9 nm corresponds clearly only to the tail tube, see Kostyuchenko et al, Nature Struct & Molec Biol, 2005). Thus, there might be a mutation in the sheath protein that prevents it to assemble in the tail. The resulting phage particle would still be produced and released from the bacteria if all other proteins necessary for cell lysis are present, but the phage particle would be defective. This would be an additional explanation so as to why no lysis is observed with the purified phage particle.

The authors also have an interesting discussion about the putative location of an endolysin-holin pair, allowing the lysis of the cell at the end of the lytic cycle. However, they identify the endolysin as a tail protein T4-gp5-like. This protein cannot be the endolysin lysing the cell at the end of the lytic cycle, because in this protein, the lysozyme active site is masked in the assemble virion. The protein gp5 is released from the virion in the host periplasm after its recognition (Kanamaru, Nature, 2002). In T4, there are two lysozymes coded in the genome: one attached to gp5, involved in infection, the second one being involved in cell lysis (Matthews & Remington, PNAS 1974). Thus, this part of the discussion needs to be reconsidered.

Other points:

  • It would help if in figure 1, the tail proteins were labelled with the proposed function based on the data presented in Table 2. Gene organisation in the structural gene region is very conserved among phages, and it would help to be able to see the organisation of genes in KIRK. (By the way, Table 2 was not in the manuscript but presented as unpublished data, why? Also, the legend in Fig 1B is far too small. “Replication and regulation” and “phage particle production” are barely readable, and the gene numbers are truncated!) To identify protein function, did the authors consider using the powerful tool HHPRED (https://toolkit.tuebingen.mpg.de/#/tools/hhpred)? It is very easy to use and extremely powerful to find homologues based on the secondary structure prediction rather than sequence similarity. I highly recommend it.
  • Having the sequence of the different primers (Table 1) is not very helpful and could go in the supplementary information. On an other hand, a schematics of why and which primer is used to amplify either the episomal or the genomic form of the prophage, or the integration of the prophage etc, would help a lot! For example, I wondered in figure 4 why was the positive control chromosomal DNA for Fha2-3 primers when those should amplify the episomal form of the prophage?
  • The production of phage particles from the KIRKrec leads to capsids only (not “similar phage particles as KIRK”, please change!). I am not completely surprised by this result: the addition of the kanamycin resistant gene will have increased the size of the phage genome, which might not be able to be entirely packaged into the capsid anymore? Anyway, with capsid-only particles, it is obvious that these latter are not infectious, for the reasons described in the discussion.
  • The two first paragraphs of the discussion would be more appropriate in the introduction as it helps understanding the genesis of the project.
  • P8, lines 347-349: “In a second stage, long linear concatemers are synthesized by rolling circle replication and the concatemers are cut at the cos-sites (cohesive ends) into virus-sized length by the terminaseand transported into the head structure by the holin (portal) protein [39].”  The terminase is bound to the portal protein, so that the two events (transport and cutting) are done concomitantly. The portal is NOT a holin!

Minor points:

  • The text within all the figures is written far too small. It is hardly readable. Please write bigger.
  • Please reconsider punctuation in the following sentences: “In individuals with compromised immune systemopportunistic infections by other Francisella species such as novicida (Fno), F. hispaniensis (Fhi) and F. philomiragia (Fph) have been reported [4-6].”  “After replication and synthesis of new virion particles, are released by lysing and consequently killing their host cells.”  “After 30 min of curing 10 μL aliquots of purified phage samples (see 2.5) were pipetted on the soft-agar,”
  • “Phage particles could be produced, but not in a way to constitute a functional bacteriophage [11].” Not clear.
  • P8, line 361: “On the other hand, the obtained tail structure is comparatively short (82 nm) for Siphoviridae ” Do you have a reference for that ?
  • P8, lines 380-385 explaining the mode of action od endolysins and holins is maybe a bit long…
  • “In sum” => “In summary”?
  • Ref 1 does not have a title.
  • The legend of panel G of figure 4 is very confusing and does not seem to correspond to the figure shown… or rather the figure is not legend accordingly (7-12?). If sizes are given in the legend, please label the ladder in panel G.

Author Response

The paper from Köppen et al describes the thorough characterisation of a prophage that can be induced to form bacteriophage particles that are released by UV light, from a Francisella bacterial strain. This is the first phage described for these bacterial species, opening the way to using it, after optimisation, as a means to fight against the pathogenis species F. tularensis.
This work is clearly important. The experiments are well planned, performed carefully and well illustrated. The authors cite the literature adequately, and give credit to related research.
However, I have a few concerns about the interpretation of some of the experiments.

R:Thank you for the kind words and the helpful suggestions to optimize the manuscript.

The authors conclude that phage KIRK, when induced by UV irradiation, is a biological active phage. I would not be so conclusive, as it could not be shown that the particle could indeed infect, yet induce lysis of bacteria. The authors very rightly conclude that KIRK might be a myophage, but the EM images do not support this idea, the tail being too thin. I completely support this conclusion, and would go further in the analysis: from the genetic/phylogenetic analysis, the prophage seems to be coding for a contractile tail. The produced particles clearly lack the sheath around the tail tube (a diameter of 9 nm corresponds
clearly only to the tail tube, see Kostyuchenko et al, Nature Struct & Molec Biol, 2005). Thus, there might be a mutation in the sheath protein that prevents it to assemble in the tail. The resulting phage particle would still be produced and released from the bacteria if all other proteins necessary for cell lysis are present, but the phage particle would be defective. This would be an additional explanation so as to why no lysis is observed with the purified phage
particle.

R: We did some additional infection experiments (see lines 247 to 250 and 376 to 378) and yet could not show an infection of bacteria by KIRK. We agree with the reviewer and we now stated that the version of KIRK present in Fhi 3523 may be a defective phage because of the lack of the tail sheat (lines 436 to 440). [given lines are line numbers of the uploaded word document]

The authors also have an interesting discussion about the putative location of an endolysinholin pair, allowing the lysis of the cell at the end of the lytic cycle. However, they identify the endolysin as a tail protein T4-gp5-like. This protein cannot be the endolysin lysing the cell at the end of the lytic cycle, because in this protein, the lysozyme active site is masked in the assemble virion. The protein gp5 is released from the virion in the host periplasm after its recognition (Kanamaru, Nature, 2002). In T4, there are two lysozymes coded in the genome:
one attached to gp5, involved in infection, the second one being involved in cell lysis (Matthews & Remington, PNAS 1974). Thus, this part of the discussion needs to be reconsidered.

R: Thank you for this correction. We reorganized this section (lines 540 ff)

Other points:
• It would help if in figure 1, the tail proteins were labelled with the proposed function based on the data presented in Table 2. Gene organisation in the structural gene region is very conserved among phages, and it would help to be able to see the organisation of genes in KIRK. (By the way, Table 2 was not in the manuscript but presented as unpublished data, why? Also, the legend in Fig 1B is far too small. “Replication and regulation” and “phage particle production” are barely readable, and the gene numbers are truncated!) To identify protein function, did the authors consider using the powerful tool HHPRED (https://toolkit.tuebingen.mpg.de/#/tools/hhpred)? It is very easy to use and extremely powerful to find homologues based on the secondary
structure prediction rather than sequence similarity. I highly recommend it.

R: We corrected and changed Figure 1 as suggested by the reviewer. Also the location of used primers are now indicated and Tab.1 is now part of the supplemental material (see comment below). Table 2 (now Tab. 1) should be part of the publication, this was an “upload mistake”. In all figures the font
size has been changed.
We used the suggested toolkit and the results are given in Tab. 1

• Having the sequence of the different primers (Table 1) is not very helpful and could go in the supplementary information. On an other hand, a schematics of why and which primer is used to amplify either the episomal or the genomic form of the prophage, or the integration of the prophage etc, would help a lot! For example, I wondered in figure 4 why was the positive control chromosomal DNA for Fha2-3 primers when those should amplify the episomal form of the prophage?

R: Primers, see above. The figure legend of Fig. 4 has been changed and “control DNA” in the different parts of the figures has been specified (line 679; within the text body kines 319 to 323 (154 to 158).

• The production of phage particles from the KIRKrec leads to capsids only (not
“similar phage particles as KIRK”, please change!). I am not completely surprised by this result: the addition of the kanamycin resistant gene will have increased the size of the phage genome, which might not be able to be entirely packaged into the capsid anymore? Anyway, with capsid-only particles, it is obvious that these latter are not infectious, for the reasons described in the discussion.
The sentence has been changed. We agree with the reviewer; we also have discussed the conclusion of a bigger head due to the additional KmR and gfp genes, but we deleted it from the manuscript. It is now mentioned at lines 398 to 400.

• The two first paragraphs of the discussion would be more appropriate in the
introduction as it helps understanding the genesis of the project.

It has been changed according the suggestion of the reviewer

• P8, lines 347-349: “In a second stage, long linear concatemers are synthesized by rolling circle replication and the concatemers are cut at the cos-sites (cohesive ends) into virus-sized length by the terminaseand transported into the head structure by the holin (portal) protein [39].” The terminase is bound to the portal protein, so that the two events (transport and cutting) are done concomitantly. The portal is NOT a holin!

Thank you for the correction, the paragraph has been changed.

Minor points:
• The text within all the figures is written far too small. It is hardly readable. Please write bigger. (It has been changed, see above)
• Please reconsider punctuation in the following sentences: “In individuals with
compromised immune systemopportunistic infections by other Francisella species such as novicida (Fno), F. hispaniensis (Fhi) and F. philomiragia (Fph) have been reported [4-6].” “After replication and synthesis of new virion particles, are released by lysing and consequently killing their host cells.” “After 30 min of curing 10 μL aliquots of purified phage samples (see 2.5) were pipetted on the soft-agar,” (The sentences have been corrected)
• “Phage particles could be produced, but not in a way to constitute a functional
bacteriophage [11].” Not clear. (It has been corrected)
• P8, line 361: “On the other hand, the obtained tail structure is comparatively short (82 nm) for Siphoviridae” Do you have a reference for that ? (The sentences has been changed and a reference is given (lines 440 to 441)
• P8, lines 380-385 explaining the mode of action od endolysins and holins is maybe a bit long…(It has been shortened)
• “In sum” => “In summary”? (It has been corrected)
• Ref 1 does not have a title. (The title is ”Tularemia“)
• The legend of panel G of figure 4 is very confusing and does not seem to correspond to the figure shown… or rather the figure is not legend accordingly (7-12?). If sizes are given in the legend, please label the ladder in panel G. (It has been changed, see above)